# Investigation of the Effect of Particle Surface Charge and Dispersion Stability on Latex Behavior in Cement Using Non-Ionic and Traditional Latexes

**DOI:** 10.3390/ma15176145

**Published:** 2022-09-05

**Authors:** Dongliang Zhou, Han Yan, Yong Yang, Xin Shu, Lei Chen, Changcheng Li, Qianping Ran

**Affiliations:** 1School of Material Science and Engineering, Southeast University, Nanjing 211189, China; 2State Key Laboratory of High Performance Civil Engineering Materials, Nanjing 211103, China

**Keywords:** nano latex, cement, dispersion stability, strength, adsorption

## Abstract

In this work, a novel total non-ionic polystyrene-polyurethane (PS-PU) composite latex was synthesized with polymerizable polyethylene glycol ether. Contrary to traditional styrene-butyl acrylate latex (St-BA), PS-PU has a smaller size and superior dispersion stability, and it is stable in saturated Ca(OH)_2_ even after 72 h. In fresh-mixed mortars, PS-PU showed a little adverse effect on workability and insignificant air entrainment, with little defoamer consumption. The retardation effect of PS-PU is also much milder than traditional St-BA. As for strength, PS-PU showed a less adverse effect on early and late age compressive strength, but its effect on flexural strength is not as pronounced as St-BA at high dosages (4% and 6%). The different behavior in cementitious materials between PS-PU and St-BA can be reasoned from their different adsorption behavior and surface charge properties, as the results from characterizations suggest. The non-ionic nature of PS-PU made it less prone to destabilization and adsorption, which turned out as the aforementioned behavior in cementitious systems. The difference can further be ascribed to the difference in their polymeric structure and properties.

## 1. Introduction

Cementitious material is the core material among building materials [1]. Brittleness is cementitious material’s major problem since it is an inorganic material [2]. Multiple methods have been employed to solve this problem, such as fiber modification [3], mix design [4] and polymer modification [5,6,7,8,9]. Among these techniques, polymer-modified cement/concrete (PMC), which applies polymer latexes in cementitious material modification by mimicking the structure of biominerals [10,11], has gained major research concern in the past decades [12,13,14,15,16].

However, there are stills some limitations to PMC, such as high dosage (10–30% binder mass), compressive strength loss and excessive air entrainment [17]. Compressive strength is one of the key indexes of cementitious materials, which directly relates to the reliability of cementitious segments in building structures [18]. Despite the relative gain in flexural strength, PMC frequently causes considerable, even severe, loss in compressive strength. The loss in compressive strength is largely due to the inhibition of cement hydration by latex, which is further caused by the adsorption of charge-rich latex particles on clinkers [19,20]. Undesired air entrainment that far exceeds adequate values if no proper defoaming is conducted is another adverse effect of polymer latexes, which not only deteriorates strength but also causes workability problems [21,22]. Excessive air entrainment usually requires a considerable amount of defoamer to mitigate [23]. Considering the above problems, PMC is becoming less popular in the past decade. The adverse effect of PMC is largely caused by the conventional preparation technique, emulsion polymerization [24]. In a typical emulsion polymerization process, a large number of ionic surfactants were involved, which makes the resultant latex particle charge rich on the surface, thus prone to adsorb on clinkers in early hydration and causing hydration inhibition. Additionally, free surfactants by excess dose in the preparation and release from latex particles are the main cause of air entrainment. Moreover, the size of conventional polymer latexes (millimeter scale) is difficult to fit in fine structures of hydration products, which causes further problems in polymer-cement compatibility [25,26].

Fortunately, with the advance in nanotechnology and new polymer design techniques, new solutions have been found to solve the above problems [27,28,29,30,31]. Due to their nano-scale size, nano latexes are able to achieve finer structures [15,28] in the hydration product network, which promotes compatibility. Additionally, surface structure and modification of nano materials are also readily tailorable, making them versatile [32,33]. The incorporation of nanotechnology in cement-specified latexes may help solve the aforementioned problems.

In this work, a novel polymer-polyurethane composite latex was prepared. The composite latex uses a novel non-ionic, polymerizable polyethylene glycol as the surfactant and stabilizing polymer sector, and the sector was incorporated into the polymeric network by polyurethane structure. Unlike traditional polymeric latex or polyurethane, no ionic surfactant was involved in the preparation, and the polymerizable surfactant was covalently bound to the polymer network and thus not easy to release. The latex’s stability and its effect on cement hydration and strength were studied, and a comparison with conventional latexes was also conducted.

## 2. Materials and Methods

### 2.1. Materials

The nano latex was prepared by a two-step condensation-emulsion polymerization.

Monomers for preparation of the latexes are as follows: styrene (A.R., Sinopharm Co., Beijing, China); toluene diisocyanate (TDI, A.R. Sinopharm Co.); polypropylene glycol (PPG, Mw = 4000, Sinopharm Co.); methallyl polyethylene glycol (HPEG, Mw = 1200, Nanjing Bote New Materials Co., Ltd., Nanjing, China); styrene (A.R., Sinopharm Co.); butyl acrylate (A.R. Sinopharm Co.); sodium dodecyl benzene sulfonate (SDBS, C.P. Sinopharm Co.), 2-Acrylamide-2-methylpropanesulfonic acid (AMPS, C.P. Sinopharm Co.) and Triton X-100 (C.P., Sinopharm Co.)

Dibutyl tin dilaurate was used as the catalyst for condensation. Initiators for polymerization were sodium hydrogen sulfite and potassium persulfate.

The cement used in this study was a P I 42.5 one (based on GB-8076-2008 [34]). Contents of the cement are in Table 1. The super plasticizer (PCA-VIII, polycarboxylate-type) and the defoamer (PXP-1, silicone-based) used in this study were supplied by Jiangsu Sobute New Materials Co., Ltd. (Nanjing, China) Other supplementary regents for characterizations and tests, including isopropanol (A.R.) and calcium hydroxide (Ca(OH)_2_/CH, A.R.), were also purchased from Sinopharm Co.

### 2.2. Preparation and Characterization of the Polyurethane-Polystyrene Latex

#### 2.2.1. Preparation of the Latex

The latex was prepared in 2 steps: preparation of an amphiphilic polyurethane block macromonomer and subsequent emulsion polymerization using the macromonomer as a polymer building block and surfactant.

PPG and HPEG were firstly pretreated to remove water residuals: they were vacuum (10 mmHg) dried at 100 °C for 24 h. Styrene was vacuum distilled (3–5 kPa) to remove inhibitors.

Condensation of the amphiphilic polyurethane macromonomer was conducted in a dry environment. Firstly, 40 g of PPG-4000 was added to a flask and cooled to below 20 °C. Then, 3.80 g of TDI was added to the flask and stirred for 10 min, 0.02 g of dibutyl tin dilaurate was added to the flask, and the system was then heated to 50 °C and stirred for 6 h. Afterward, 28.4 g of HPEG was added dropwise to the flask in 10 min, and HPEG was pre-heated to 60–65 °C to maintain its liquid form. The system was kept at 50 °C under stirring for another 6 h.

After the condensation, the macromonomer (60 g) was poured into a beaker and cooled to below 20 °C; the resultant solid was broken into pieces and dissolved in 40 g of styrene at 10–20 °C. The viscous solution was then dispersed in 500 mL of distilled water at 600 rpm for 30 min. Upon full dispersion, 0.85 g of potassium persulfate was dissolved in the monomer dispersion; then, the dispersion was heated to 55 °C, purged with N_2_, and 50 mL of a solution containing 0.11 g of sodium hydrogen sulfite was added to by a peristaltic pump at a rate of 0.33 mL/min, after the addition, the system was kept at 55 °C for another 0.5 h. Finally, unreacted monomers were removed by vacuum (3–5 kPa at 30 °C for 2 h) and the resultant latex was stored for further use. The chemical route for preparation is demonstrated in Figure 1.

A reference styrene-butylacrylate latex (noted as St-BA in the following Appendix A) was also prepared; the detailed procedure is available in Appendix A.

#### 2.2.2. Characterization of the Latex

After preparation, the solid content of the nano latexes was firstly measured, and the conversion rate was roughly estimated. Dispersion stability of the latex was verified using saturated Ca(OH)_2_ solution to simulate a pore solution environment; the latexes were diluted into 0.1% dispersions for better observation. The size of the particles in the latexes in different environments was measured by Dynamic Light Scattering (DLS, Type CGS-3, ALV Co., Langen, Germany); samples were prepared as 0.05 % (*w*/*w*) dispersion with ultrapure water and saturated Ca(OH)_2_ solution. Morphology of the latexes was verified by SEM (FEI Quanta 250, 15 kV, 50,000× *g*); the latexes were also sampled at 0.05% to inhibit membrane formation. Zeta potential of the latex dispersions was measured by a DT-300 (Dispersion Technology Inc., New York, NY, USA) zetaprobe at the intrinsic pH of the as-prepared latex. Fourier transform infrared spectra (FT-IR) of the samples was acquired on a FT-IR spectrometer (Type Nicolet 370, Thermo Fisher Co., Waltham, MA, USA).

### 2.3. Mortar Testing

The mortars in this study were prepared based on the procedures of GB/T 17671-1999 [35]. The w/b ratio (w/b) was 0.4, and the binder/sand ratio was 1:2.7. Mix design of latexes and references in cement composites was presented in Table 2. The flow of the mortars was regulated to 160 ± 5 mm. After mix, fluidity and density were measured, then supplementary defoamer (0.02–0.10 g) was added for latex-added mortars, and the mortar was remixed at a high stirring rate for 15 s. The process was repeated until the density plateaued, which is necessary to avoid disturbance from excessive air in subsequent tests.

The mortars were then cast and cured at 20 ± 1 °C and 95% relative humidity. Three batches of mortar prisms were prepared for ages of 1 d, 7 d and 28 d; flexural and compressive strength of mortar was tested afterward.

### 2.4. Paste Characterization

#### 2.4.1. Paste Preparation

Unless specifically noted, the pastes were prepared according to GB/T 8077-2012 [36] at a w/c of 0.4. The pastes here were prepared with cement replaced by 2% and 4% latex samples and appropriate amount of PCA-VIII and PXP-I to regulate the flow (200 ± 5 mm) and air entrainment (with blank as reference). The admixtures were added to the water phase prior to mixing with cement.

Setting time of cement paste, which characterizes the point of early hydration product network formation, reflects the rate of early cement hydration and further indicates the impact of cement admixtures. The setting time of the pates was measured according to the procedures in GB/T 1346-2011 [37].

#### 2.4.2. Characterization of Early Age Hydration

Hydration heat evolution in early ages was characterized by isothermal calorimetry (IC). In the tests, about 13.8 g of the paste (prepared according to Section 2.4.1) was accurately weighed into a plastic vial. The vial was sealed and placed in a TAM Air isothermal calorimeter to measure heat development for 24 h at 20.0 °C.

Zeta potential of latex-modified pastes was measured using the DT-300 zetaprobe, as described in Section 2.2.2. The pastes were prepared by the procedure in Section 2.4.1 and were directly measured.

Adsorption of latex on cement particles at the start of hydration was characterized by assessing the remaining latex in the supernatant, which is measured by Total Organic Carbon (TOC) analysis. For a more convenient extract of supernatant, the w/c here is 1.0. In the experiment, 50 g of the cement with 1.0%, 2.0%, 4.0% and 6.0% latex addition were mixed with 50 g of water; the mixture then underwent the same procedures as Section 2.4.1. after mixing, the paste was centrifuged at 3000 rpm for 10 min (as pre-tested, this setting is sufficient to separate cement while keeping latex particles in the supernatant), and the supernatant was collected, diluted and tested on a Multi N/C 3100 TOC analyzer (Analytik Jena, Jena, Germany).

#### 2.4.3. Characterization of Hydration at Later Ages

In the experiments, 25 g of paste was cast in a 50 mL plastic vial and sealed thereafter. After the target curing times (7 d, 28 d) at 20 °C, samples were demolded. The outer layer (1 mm of thickness) was removed. Samples for SEM scans were split into lumps of 3–5 mm, and hydration was suspended by 24 h isopropanol (A.R.) treating for 3 cycles, after which the samples were dried in a vacuum at 30 °C and sealed under N_2_. Samples used for XRD and TGA test were ground and treated with isopropanol in the above process. After treatment, the sample powders were collected, sieved (180-mesh), vacuum dried (30 °C), and sealed in N_2_-filled tubes.

SEM observations were also conducted in an FEI Scanning electron microscope (Type Quanta 250, FEI Co., Hillsboro, OR, USA), with an acceleration voltage of 15 kV. A X-ray diffractometer was used for XRD (Type D8 Advance, Bruker Co., Rheinstetten, Germany). Before testing, 10% of α-Al_2_O_3_ was introduced as internal reference. The spectra were analyzed using the Rietveld method that had been pre-installed in TOPAS.

## 3. Results

### 3.1. Preparation and Characteristics of Polymeric Nanoparticles

Physiochemical properties of the latexes are shown in Table 2. According to Table 2, the conversion rate of both latexes is higher than 90%, which confirms successful preparation. Then, surface properties of the latex particles were assessed by zeta potential analysis. The data of the latexes at their intrinsic pH are also presented in Table 3. Unlike traditional latexes, the zeta potential of PS-PU (−5.2) is close to 0, which can be attributed to its non-ionic nature, while St-BA is −27.3 mV. These data confirmed the non-ionic nature of PS-PU.

Figure 2a shows the dispersion of the latexes in different media; as can be observed, PS-PU remained stable in Ca(OH)_2_ solution after 72 h, while segregation had begun in St-BA’s Ca(OH)_2_ solution at a ring of precipitate appeared around the surface. The mean hydraulic radius (R_h_) of the latexes is 40.3 nm in water, which is small compared with traditional latexes (50–100 nm in water) [5,13]; the R_h_ of St-BA latex (~80 nm) also fell in this range. In the Ca(OH)_2_ solution, the difference between the two latexes became more distinct, as Rh of PS-PU only increased to 75.3 nm while Rh of St-BA had more than quadrupled. DLS data clearly showed the superior dispersion stability of PS-PU.

SEM images of the latexes are shown in Figure 2b. As can be observed, PS-PU was presented as a free particle and small clusters, while St-BA formed stripe-like membranes. The size of the remaining particles of St-BA was also larger.

Characteristic peaks of polymeric segment in PS-PU and St-BA, including aromatic, alkyl (backbone in St-BA and PEG/PPG in PS-PU) and carbonyl groups, can be found in FT-IR spectra of the samples in Figure 2c. The intensity of the characteristic peaks showed considerable deviation, which is mainly due to the difference in abundance between the two materials. The much smaller amount of aromatic and carbonyl groups in PS-PU (from TDI segments), as compared with those in St-BA (from phenyl group in St and carboxylate group in BA, respectively), resulted in low peak intensity.

Characterizations of the latexes suggested the non-ionic nature of PS-PU, which may inhibit its adsorption on cement particles by electrostatic force. The inhibition, in turn, ensured its stability in the pore solution environment. Additionally, the covalent binding of the surfactant groups (PEG) in PS-PU can further improve its stability by avoiding destabilization caused by surfactant desorption.

### 3.2. Effect of the Nano Latexes on Mortars

#### 3.2.1. Mortar Fresh Properties

The effect of the nano latexes on fresh mortar is shown in Table 4. As the results suggest, the addition of PS-PU slightly improved the fluidity of the fresh mortars, and the improvement increased with PS-PU dosage. The improvement is due to the relative decrease in binder content and water-reducing effect from PS-PU, which is especially prominent at high dosage. The water-reducing effect is likely due to its structure, i.e., polymer core with surface PEG chain, which resembles polycarboxylate cement dispersant to some extent. Steric hindrance effects from the PEG chain on PS-PU may be the main contributing factor to its water-reducing effect. Additionally, previous research has reported water reduces the effect of PEG-modified nano polymer latex [38].

Compared with PS-PU, St-BA latex also improved the fluidity of the mortars, which is based on the same reason, but the degree of improvement is weaker, which may be due to adsorption of the latex particles by cement and destabilization caused by surfactant desorption.

Excessive air entrainment that is caused by surfactants in the latex has always been a major drawback of PMC; a considerable amount of defoamers is required to mitigate the effect. The air-entraining ability of the latexes was characterized by the density of the mortars without defoamer addition and the number of defoamers to drive the density of the samples as close to the blank.

As the results in Table 4 suggest, the air entraining effect of PS-PU is unconventionally low; compared with St-BA latex, its density loss is insignificant (from 2.29 × 10^3^ kg m^−3^ to 2.28 and 2.27 × 10^3^ kg m^−3^, respectively) at low dosages (1%, 2%), and not pronounced at high dosages (4%, 6%), which is only 24–37% of St-BA latexes’ value before defoaming. The amount of defoamers to mitigate the effect is also much lower, which is 13–18% of St-BA latex.

#### 3.2.2. Mortar Strength

The strength of the mortar samples with 1–6% latex addition at 1–28 d is shown in Figure 3. As the results suggest, both the latexes showed retarding effect at an early age (1 d), yet the effect of ST-BA is more pronounced, with 1 d compressive strength decreased by 27–65% from 1–6% dosage, while the strength decrease in PS-PU is only 18–41% at the same dosage. The strong retarding effect of St-BA is due to the hydrolyzation of the acrylic esters within, which exposed the carboxyl groups in the polymer chain and enhanced its adsorption on cement particles. The stronger adsorption further resulted in retardation. Compared with St-BA, the lower strength loss of PU-PS modified mortars can be attributed to its non-ionic and non-hydrolyzable structure, and the decrease is mainly caused by the relative decrease in binder content. Compared with compressive strength, flexural strength decrease in the samples was relatively smaller, which is characteristic for cementitious materials modified by polymer latexes. The less flexural strength loss and thus higher flexural/compressive strength ratio is due to the formation of a composite organic/inorganic network made up of polymer and hydration products.

As for later ages, the compressive strength loss of the mortars gradually lessened. In the two latexes, the strength loss of St-BA is still more significant, which is 13–48% at 7 d and 8–31% at 28 d. The strength distribution of the samples at 28 d is interesting: flexural strength decrease in St-BA is pronounced at low dosage (1%, 2%) but lessened at high dosages (4%, 6%), while flexural strength loss of PS-PU is insignificant at the low dosages, but turned prominent at the high dosages. This results in PS-PU’s high flexural-compressive ratio at the low dosage and St-BA at the high dosage. The different flexural strength variation may be due to the difference in dispersion stability and film-forming between the two latexes: at low dosages, St-BA was unable to form a film network throughout the binders, and its adverse effect on strength was not mitigated by the formation of organic/inorganic network. PS-PU is more stable and affects less on strength at these dosages; while at high dosages, the network of St-BA can be formed, but the film of PS-PU is still hard to form due to its stability. As for compressive strength, the decrease is still smaller for PS-PU due to its stability.

### 3.3. Interaction between the Latexes and Cement

#### 3.3.1. Interaction in Early Ages

Setting times of the paste are listed in Table 5; the results confirmed the retarding effect of the latexes, the initial setting of pastes with PS-PU addition is 260 min and 350 min, respectively, which is postponed by 70 and 160 min, and the final setting was further delayed by 80 and 220 min. Compared with PS-PU, the setting time delay of St-BA is even more serious, as the initial set at 2% dosage is already 360 min, the time turned 520 min at 4% dosage, but the time interval between the initial and final set is relatively narrower.

IC curves of the latexes are presented in Figure 4a. The curves further confirmed the retarding effect, which was found in strength tests. The main peaks of both latexes were postponed, and the degree of delay rose with dosage. Additionally, setting times that were deduced from the curves were in agreement with results from direct measurement on pastes.

Early age interaction between the latexes and cement particles were assessed by zeta potential and adsorption tests, and the results are demonstrated in Figure 4b,c. As zeta potential results suggest, the variation tendency of pastes with latex addition in the first 30 min is different for PS-PU and St-BA: zeta potential values of St-BA modified pastes were far more negative than those with PS-PU modification. The zeta potential of the paste with 2% PS-PU modification dropped from 14.7 to 13.9 at 4 min and 17.3 to 16.6 at 30 min, while the value of the paste 2% St-BA was from 14.7 to 11.7 at 4 min. At 4% dosage, the zeta potential change of PS-PU modified cement was still close to those at 2% dosage, while an increase from 2% St-BA to 4% resulted in a significant zeta potential decrease. The different trend in zeta potential variation may be due to the difference in ionic properties, which further lead to different adsorption affinity with cement particles. This assumption was verified by adsorption assessment by TOC. As the data in Figure 4c suggest, the adsorption capacity of PS-PU by cement is much lower than those with St-BA, only 40–63% the amount of the latter at 2 h, and the adsorption was less affected by dosage and time, with a much lower increase by dosage and a milder time-elapse increase. The lower adsorption of PS-PU is apparently due to its non-ionic and non-hydrolyzable properties. These data confirmed the previous suggestion.

#### 3.3.2. Interaction in Later Ages

SEM images of the paste at ages of 7 d and 28 d are shown in Figure 5. As the images suggest, the difference between the samples was not very significant; there seem to be fewer pores in PS-PU and St-BA modified samples, especially in PS-PU modified samples. No clear film or membrane formation was observed in the images, which may be due to the relatively low dosage of the latexes.

XRD spectra of the samples at 7 d and 28 d are demonstrated in Figure 6a,b. Content of C_3_S, C_2_S and portlandite (i.e., the usual crystalline form of Ca(OH)_2_ as cement hydration product) are also shown in Figure 6c. To evaluate the degree of hydration, the ratio between portlandite and C_3_S/C_2_S (noted as CxS in the following Appendix A) was calculated. As the data suggest, the degree of hydration was inhibited by the addition of both latexes, as the CH/CxS fell from 0.94 in blank by 5–15% in the PS-PU modified samples and 20–22 in St-BA modified samples. At 28 d, the decrease is 11–15% for PS-PU and 15–28% for St-BA. Still, the decrease in portlandite content for PS-PU is lower than that of St-BA, indicating a weaker hydration inhibition at later ages, which may be due to PS-PU’s less film forming.

## 4. Discussion

As the above data suggest, the latexes exhibited distinctly different impacts on fresh properties of mortar, hydration and strength. The difference may arise from the dispersion stability of the latexes.

As the characterizations in Section 3.1 suggest, PS-PU is highly stable in a cementitious environment due to its non-ionic nature, while St-BA would gradually lose its stability due to desorption of surfactants and hydrolyzation of the ester groups. The superior stability of PS-PU made it less likely to destabilize and adsorb on cement particles, which in turn exhibit its milder impact on workability. Additionally, the surfactant groups on PS-PU are highly unlikely to unbound (instead of desorb as it is covalently bounded) and transfer onto cement particles, which almost eliminated its air-entraining effect. The less affinity for cement also led to weaker hydration inhibition, as the cement particles were less hindered by adsorbed latex particles or surfactants from the latex, as results from Section 3.2 and Section 3.3.1 suggest. The mechanism for PS-PU and St-BA’s different impacts on cement workability and strength is demonstrated in Figure 7.

Finally, St-BA’s instability still bears some advantages, which brought about the ease for film in binder network at later ages, which is beneficial for flexural strength. PS-PU’s superior stability may also inhibit its effect on flexural strength improvement, as the results from Section 3.2 suggest.

## 5. Conclusions

In this work, a novel non-ionic, non-hydrolyzable polystyrene-polyurethane (PS-PU) composite latex was prepared using polymerizable polyethylene glycol ether.

Compared with traditional styrene-butyl acrylate latex (St-BA), PS-PU is smaller in size and exhibits superior dispersion stability, and it is stable in saturated Ca(OH)_2_ even after 72 h. In fresh-mixed mortars, PS-PU showed a little adverse effect on workability and insignificant air entrainment, with little defoamer consumption. The retardation effect of PS-PU is also much milder than traditional St-BA. As for strength, PS-PU showed a less adverse effect on early and late age compressive strength, but its effect on flexural strength is not as pronounced as St-BA at high dosages (4%, 6%). The different behavior in cementitious materials between PS-PU and St-BA can be reasoned from their different adsorption behavior and surface charge properties, as the results from characterizations suggest. The non-ionic and non-hydrolyzable nature of PS-PU made it less prone to destabilization and adsorption, which turned out as the aforementioned behavior in cementitious systems. The difference can further be ascribed to the difference in their polymeric structure and properties.

In summary, the results in this study suggest that the stability of the latexes can greatly affect their effect on the workability and strength of cement.

## Figures and Tables

**Figure 1 materials-15-06145-f001:**
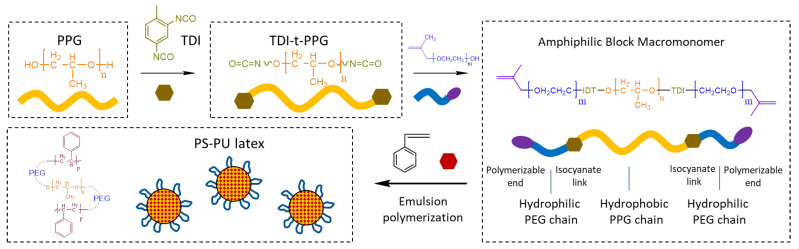
Chemical route for the preparation of PS-PU.

**Figure 2 materials-15-06145-f002:**
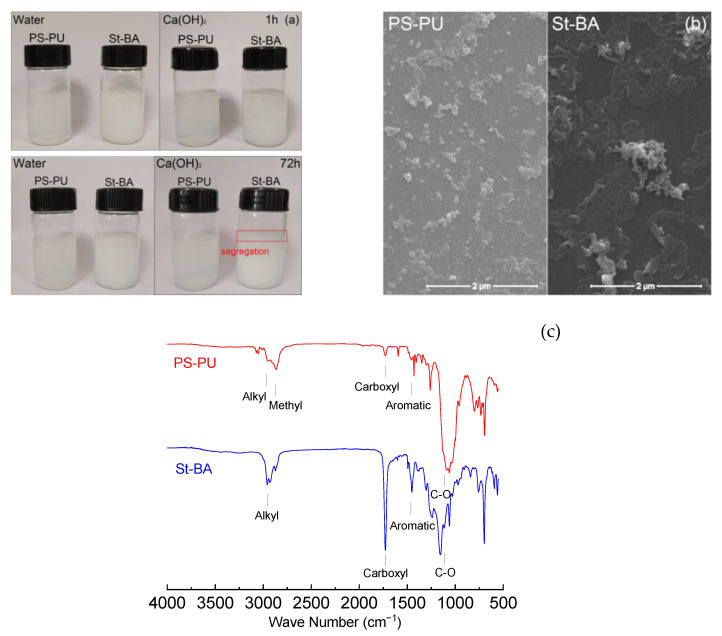
Dispersion status of the latex samples (**a**), SEM images of the samples (**b**) and FT-IR spectra of the samples (**c**).

**Figure 3 materials-15-06145-f003:**
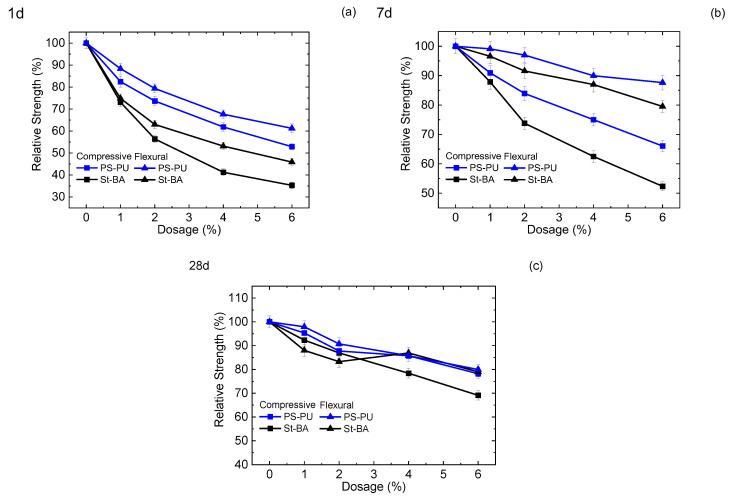
Strength of cement mortars with latex modification: (**a**) 1 d; (**b**) 7 d; (**c**) 28 d.

**Figure 4 materials-15-06145-f004:**
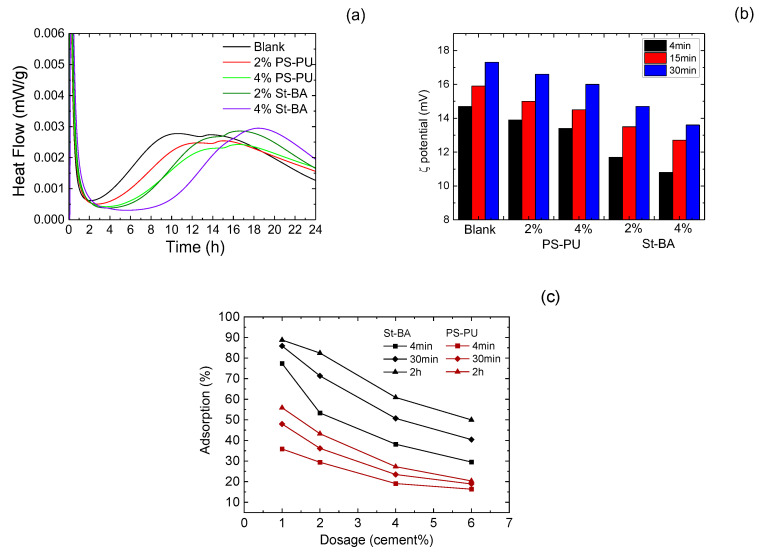
Interaction of the latexes between cement in early ages: (**a**) hydration heat evolution; (**b**) zeta potential evolution; (**c**) adsorption of the latexes on cement in 4 min–2 h.

**Figure 5 materials-15-06145-f005:**
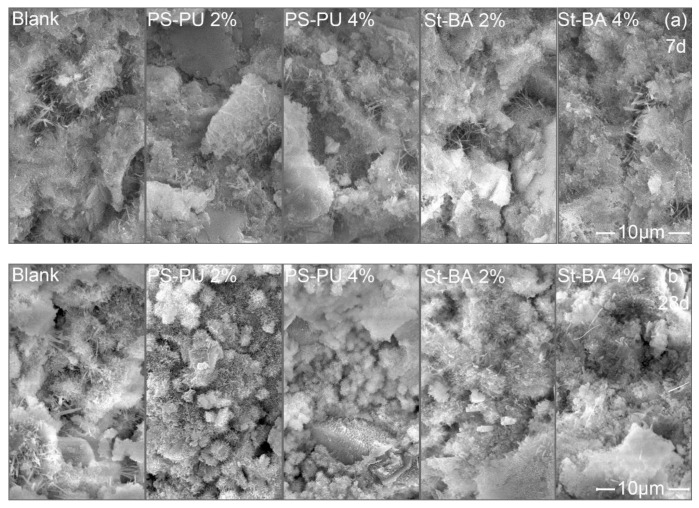
SEM images of hardened paste samples with latex modification: (**a**) 7 d and (**b**) 28 d. The scale bar applies to all the sub-images.

**Figure 6 materials-15-06145-f006:**
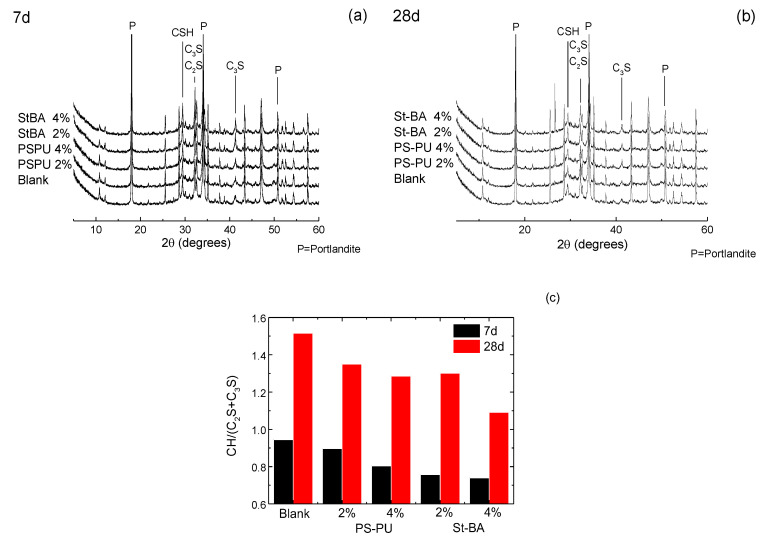
XRD characterizations of the latex-modified mortars: (**a**) XRD spectra at 7 d; (**b**) XRD spectra at 28 d; (**c**) CH/(C_2_S+C_3_S) data based on Rietveld calculation on XRD spectra; peak identification was based on [39,40].

**Figure 7 materials-15-06145-f007:**
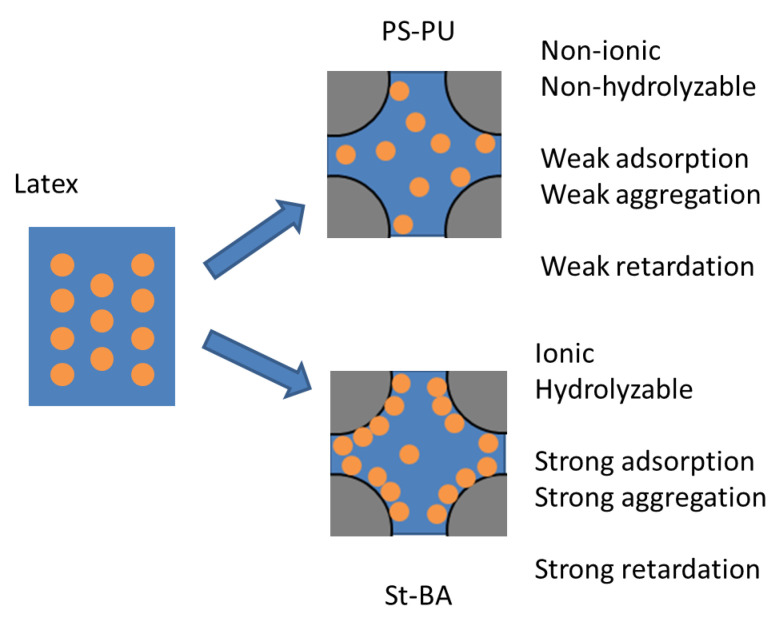
Mechanism for PS-PU and St-BA’s different impacts on cement workability and strength.

**Table 1 materials-15-06145-t001:** Composition and properties of the P I 42.5 cement.

Compound	Content (%)
SiO_2_	21.23
Al_2_O_3_	4.83
CaO	64.34
MgO	1.81
Fe_2_O_3_	3.12
SO_3_	3.32
K_2_O	0.68
Na_2_O	0.19
Free lime	1.15
Total	99.26

**Table 2 materials-15-06145-t002:** Mix design of the latexes and references in cement composites.

Content	Blank	Latex-Modified Samples			
		1.0%	2.0%	4.0%	6.0%
**Cement(** **k** **g** **/m^3^** **)**	560 ± 0.5				
**Water(** **k** **g** **/m^3^** **)**	224 ± 0.2				
**Sand(** **k** **g** **/m^3^** **)**	1512 ± 5				
**Latex((** **k** **g** **/m^3^** **) in solid weight)**	None	5.60 ± 0.02	11.20 ± 0.05	22.40 ± 0.10	33.60 ± 0.10

**Table 3 materials-15-06145-t003:** Physiochemical parameters of the latexes.

Samples	Solid Content (%)	Conversion (%)	Zeta Potential (Intrinsic pH)	R_h_ (water)	R_h_ (Ca(OH)_2_)
**PS-PU**	16.96	92.54	−5.2 (5.3)	40.3	75.3
**St-BA**	27.08	92.96	−27.3 (5.1)	163.7	256.2

**Table 4 materials-15-06145-t004:** Fresh properties of the latex-modified mortar samples.

Samples		Flow (mm)	Standard Deviation	Superplasticizer (% Cement)	Density before Defoaming (kg/m^3^)	Density after Defoaming (kg/m^3^)	Defoamer (g)
**Blank**		157	2.4	1.85	2.29	2.29	0.02
**PS-PU**	1.0%	162	2.6	1.53	2.27	2.27	0.02
	2.0%	163	1.7	1.13	2.24	2.27	0.04
	4.0%	159	3.6	0.67	2.19	2.24	0.04
	6.0%	160	2.6	0.03	2.12	2.21	0.04
**St-BA**	1.0%	158	2.0	1.60	2.17	2.28	0.04
	2.0%	162	2.6	1.36	1.96	2.26	0.10
	4.0%	157	1.0	0.97	1.88	2.24	0.22
	6.0%	161	1.7	0.53	1.83	2.20	0.30

**Table 5 materials-15-06145-t005:** Setting time of paste with latex modification.

Sample		Initial Set (min)	Final Set (min)
**Blank**		190	255
**PS-PU**	2%	260	355
	4%	350	475
**St-BA**	2%	360	470
	4%	520	680

## Data Availability

Not applicable.

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
