# Peer review of "Investigation of the Effect of Particle Surface Charge and Dispersion Stability on Latex Behavior in Cement Using Non-Ionic and Traditional Latexes"

_materials, 2022, doi:10.3390/ma15176145_

Round 1

Reviewer 1 Report

Zhou and co-workers present the synthesis of polystyrene-polyurethane (PS-PU) composite latex and its application as mortars additive. A comparison between cementitious materials mixed with  PS-PU and mixed with St-BA is shown and an investigation on the effect of the two different additive on the properties of the mixtures was carried out with several investigation techniques. PS-PU showed less effect on the workability, air entrainment, early and late age compressive strength but its effect on flexural strength is not as pronounced as St-BA. The different behaviour was associated to the non-ionic and non-hydrolyzable nature of PS-PU, making it less prone to destabilization and adsorption. The presented experiments are sound and a nice result comes out of this study. Hence, I think this work should be published after some revisions:

·       English language should be checked for a lot of misspelled words, missing dots, capital letters etc.

·       The introduction must be amplified, not enough information on the state of the art of these materials is given. Moreover, properties like compressive strength and air entrainment should be better explained and the effect on the final application of the cement should be enlightened.

·       In Table 2, errors should be added also for the addition of latex.

·       A brief description of setting time and the retarding effect could be added, in Par. 2.4.1 or in the introduction.

·       In Fig. 2(c), the IR spectra show the characteristic peaks of the polymers, but the intensity is quite different for the two materials, the authors could add a little more exhaustive explanation of the two spectra. Legend is also missing.

·       Line 226, do the author have an explanation of the smaller decrease of the flexural strength compared to the compressive strength?

For a better understanding to a wider audience, the meaning of the portlandite and CxS peaks in the XRD spectra should be briefly introduced (line 295). 

Author Response

Point 1: English language should be checked for a lot of misspelled words, missing dots, capital letters etc.

Response 1: We are sorry for the mistakes, misspelled words, missing dots, capital letters have been checked and revised.

Point 2: The introduction must be amplified, not enough information on the state of the art of these materials is given. Moreover, properties like compressive strength and air entrainment should be better explained and the effect on the final application of the cement should be enlightened.

Response 2: The introduction has been revised for better demonstration on state of the art of relevant field and more details about compressive strength and air entrainment has been supplied. Relevant literature citations have also been supplemented.

Point 3: In Table 2, errors should be added also for the addition of latex.

Response 3: Errors has been added in Table 2.

Point 4: A brief description of setting time and the retarding effect could be added, in Par. 2.4.1 or in the introduction.

Response 4: Relevant discussions in Par. 2.4.1 have been revised for more detailed description.

Point 5: In Fig. 2(c), the IR spectra show the characteristic peaks of the polymers, but the intensity is quite different for the two materials, the authors could add a little more exhaustive explanation of the two spectra. Legend is also missing.

Response 5: Fig. 2(c) and relevant discussion in Section 3.1 has been revised.

Point 6:  Line 226, do the author have an explanation of the smaller decrease of the flexural strength compared to the compressive strength?

Response 6: The smaller decrease and resultant higher flexura/compressive is due to the formation of organic/inorganic polymer/hydration product network, which is the major advantage of polymer-modification on cementitious materials. Relevant discussion has been revised.

Point 7:  For a better understanding to a wider audience, the meaning of the portlandite and CxS peaks in the XRD spectra should be briefly introduced (line 295). 

Response 7: Discussion in relevant section has been specified for better comprehension.

Reviewer 2 Report

More literature can be added to improve the overall background of the topic.

Author Response

Point 1: More literature can be added to improve the overall background of the topic.

Response 1: Additional literatures for extensive background comprehension have been supplemented, thanks for the reviewer's suggestion.

Reviewer 3 Report

Reviewer comments on the paper entitled “Investigation on the effect of particle surface charge and dispersion stability on latex behavior in cement using non-ionic and traditional latexes”

1.     Please correct for some typos in the text. For instance: Sompatibility (page 1, line41). I assume it is compatibility. Another one is in page 3, line 110: transfer, which should be Transform (The technique is Fourier Transform Infrared Spectroscopy. There are other typos in the whole manuscript that should be corrected after a careful revision.

2.     In table 2 please indicate the volume of mortar produced. I suggest the authors provide the mix proportions for 1 m3.

3.     In Figure 2, please provide the color code for the two spectra either in the graph or in the figure caption.

4.      In Figure 2b, the authors should include the bar scale in the left SEM image (PS-PU), so the readers can compare the size of the clusters only if the scale is the same.

5.     In Table 4, the results of fluidity for the PS-PU indicate that no superplasticizer would be required for the same flow when the 6% was added. Does this mean that its use at high dosages (6%) act as a superplasticizer? Please comment on this issue. If the experiments were done in replicate samples, please provide the standard deviation of the results obtained. In this way it is easier to see if there are clear differences or the values are similar.

6.     In the results section, when referring to the figures, please use the present tense. For instance, in page 7, line 217: …were shown in Fig. 3. Should be: are shown in Fig. 3. Then the past tense is correctly used when discussing the results.

7.     In figure 3, please provide the standard deviation of the plotted values.

8.     What is the magnification of images in figure 5?. Please provide the scale bar.

9.     In the conclusions section, page 12, line 342-344, it is not clear what they mean when say: Despite improved workability and less compressive strength loss, superior dispersion stability may have adverse effect on their performance on flexural strength improvement. I understand they are referring to the PS-PU, but in fact it behaved better than the St-BA. Please clarify this.

Author Response

Point 1: Please correct for some typos in the text. For instance: Sompatibility (page 1, line41). I assume it is compatibility. Another one is in page 3, line 110: transfer, which should be Transform (The technique is Fourier Transform Infrared Spectroscopy. There are other typos in the whole manuscript that should be corrected after a careful revision.

Response 1: We are sorry for the mistakes, the manuscript has been checked for spelling and format mistakes.

Point 2: In table 2 please indicate the volume of mortar produced. I suggest the authors provide the mix proportions for 1 m3.

Response 2: Table 2 has been revised with suggested interpretation.

Point 3: In Figure 2, please provide the color code for the two spectra either in the graph or in the figure caption.

Response 3: Fig. 2c has been revised with new legend.

Point 4: In Figure 2b, the authors should include the bar scale in the left SEM image (PS-PU), so the readers can compare the size of the clusters only if the scale is the same.

Response 4: Fig. 2b has been revised with scale bars.

Point 5:  In Table 4, the results of fluidity for the PS-PU indicate that no superplasticizer would be required for the same flow when the 6% was added. Does this mean that its use at high dosages (6%) act as a superplasticizer? Please comment on this issue. If the experiments were done in replicate samples, please provide the standard deviation of the results obtained. In this way it is easier to see if there are clear differences or the values are similar.

Response 5: The water reducing effect of PS-PU is probably due to its surface PEG chain, which acts as a polycarboxylate-like dispersant via steric effects. relevant discussions (Par. 1 in 3.2.1) and table 4 has been revised with testing errors. 

Point 6:   In the results section, when referring to the figures, please use the present tense. For instance, in page 7, line 217: …were shown in Fig. 3. Should be: are shown in Fig. 3. Then the past tense is correctly used when discussing the results.

Response 6: The grammar mistakes concerning figure referring has been corrected. thanks for the reviewer's kind suggestion.

Point 7:  In figure 3, please provide the standard deviation of the plotted values.

Response 7: Error bars of the data has been supplemented.

Point 8:  What is the magnification of images in figure 5?. Please provide the scale bar.

Response 8: Scale bar has been added in Fig. 5.

Point 9:  In the conclusions section, page 12, line 342-344, it is not clear what they mean when say: Despite improved workability and less compressive strength loss, superior dispersion stability may have adverse effect on their performance on flexural strength improvement. I understand they are referring to the PS-PU, but in fact it behaved better than the St-BA. Please clarify this.

Response 9: The confusing expression has been deleted. The original text was to express the relatively higher flexural/compressive strength ratio of St-BA modified mortar samples at high dosages, but considering the fact that this high ratio was achieved at the cost of greater compressive strength loss (as shown in Fig. 3), the overall performance of PS-PU is indeed better than St-BA on strength loss mitigation.